# Proteomic Blood Profiles Obtained by Totally Blind Biological Clustering in Stable and Exacerbated COPD Patients

**DOI:** 10.3390/cells13100866

**Published:** 2024-05-17

**Authors:** Cesar Jessé Enríquez-Rodríguez, Sergi Pascual-Guardia, Carme Casadevall, Oswaldo Antonio Caguana-Vélez, Diego Rodríguez-Chiaradia, Esther Barreiro, Joaquim Gea

**Affiliations:** 1Respiratory Medicine Department, Hospital del Mar—IMIM, 08003 Barcelona, Spain; cesarjesse.enriquez01@alumni.upf.edu (C.J.E.-R.); spascual@psmar.cat (S.P.-G.); carme.casadevall@upf.edu (C.C.); ocaguana@psmar.cat (O.A.C.-V.); darodriguez@psmar.cat (D.R.-C.); ebarreiro@imim.es (E.B.); 2MELIS Department, Universitat Pompeu Fabra, 08003 Barcelona, Spain; 3CIBERES, ISCiii, 08003 Barcelona, Spain; 4BRN, 08003 Barcelona, Spain

**Keywords:** COPD, exacerbation, proteins, inflammation, immune response, lipid profile, coagulation, complement system

## Abstract

Although Chronic Obstructive Pulmonary Disease (COPD) is highly prevalent, it is often underdiagnosed. One of the main characteristics of this heterogeneous disease is the presence of periods of acute clinical impairment (exacerbations). Obtaining blood biomarkers for either COPD as a chronic entity or its exacerbations (AECOPD) will be particularly useful for the clinical management of patients. However, most of the earlier studies have been characterized by potential biases derived from pre-existing hypotheses in one or more of their analysis steps: some studies have only targeted molecules already suggested by pre-existing knowledge, and others had initially carried out a blind search but later compared the detected biomarkers among well-predefined clinical groups. We hypothesized that a clinically blind cluster analysis on the results of a non-hypothesis-driven wide proteomic search would determine an unbiased grouping of patients, potentially reflecting their endotypes and/or clinical characteristics. To check this hypothesis, we included the plasma samples from 24 clinically stable COPD patients, 10 additional patients with AECOPD, and 10 healthy controls. The samples were analyzed through label-free liquid chromatography/tandem mass spectrometry. Subsequently, the Scikit-learn machine learning module and K-means were used for clustering the individuals based solely on their proteomic profiles. The obtained clusters were confronted with clinical groups only at the end of the entire procedure. Although our clusters were unable to differentiate stable COPD patients from healthy individuals, they segregated those patients with AECOPD from the patients in stable conditions (sensitivity 80%, specificity 79%, and global accuracy, 79.4%). Moreover, the proteins involved in the blind grouping process to identify AECOPD were associated with five biological processes: inflammation, humoral immune response, blood coagulation, modulation of lipid metabolism, and complement system pathways. Even though the present results merit an external validation, our results suggest that the present blinded approach may be useful to segregate AECOPD from stability in both the clinical setting and trials, favoring more personalized medicine and clinical research.

## 1. Introduction

Chronic Obstructive Pulmonary Disease (COPD) is a clinical entity with high prevalence (around 10% of the adults in developed countries) even though it still shows a high rate of underdiagnosis [1,2]. Moreover, COPD remains among the top three causes of mortality and the ten most frequent causes of disability-adjusted life years (DALYs) [1,2], resulting in a major health problem. This chronic disorder is mostly caused by the chronic inhalation of tobacco smoke and other hazardous chemicals, and is mainly characterized by respiratory symptoms and persistent airflow obstruction, the latter expressed as an obstructive respiratory pattern in forced spirometry [3,4,5,6,7]. Without any doubt, the need for a validated spirometry for COPD diagnosis contributes to its underdiagnosis. In addition, COPD patients also show exacerbation periods (AECOPD), defined by the acute worsening of respiratory symptoms, requiring changes in the current medication [3,4,5,6,7]. Even though this latter definition is very useful in the clinical setting, it contains enough elements of subjectivity and potential bias to make it difficult to specifically assign the ‘exacerbation’ label to a particular episode of clinical impairment. Consequently, it is critical to be easily able to identify COPD patients among the general population as well as the AECOPD episodes in such patients. Since both circumstances probably have relatively specific underlying molecular mechanisms (endotypes), these can be investigated through the search for differential biological markers and pathways. To date, two different conceptual and technical approaches have been used to cope with this challenge. The most common were those driven by specific hypotheses based on pre-existing knowledge, generally using conventional laboratory techniques [8,9,10]. However, this approach necessarily restricted the search to a relatively small list of initially preselected blood biomarkers, which can be considered as ‘the usual suspects’. Therefore, it is not surprising that the results have been relatively disappointing. The complementary approach, which has been used more recently with promising results, is to use non-hypothesis-driven and wide screening markers with omic technologies. However, once the biological signals are obtained, the most common next step has been to compare the results among predefined clinical groups, which is a ‘non-blind approach’ that potentially restricts the possible outcomes and limits their usefulness. Moreover, the majority of these latter studies have focused on specific groups of patients belonging to wide multicentric cohorts, with the advantages and disadvantages implicit in their particular inclusion criteria and a potential technical heterogeneity among the centers [9,11]. Furthermore, most of the proposed panels of biomarkers and/or models obtained with these two approaches have failed in the validation step carried out in new cohorts [12,13]. By contrast, the use of non-hypothesis-driven statistical methods (such as the clustering approach) in the second step of the analysis of omic data can provide some additional advantages for blindly classifying individuals into distinct groups, whose members will share biological features. The main objective of the present study was to test the feasibility of this approach. Therefore, we initially obtained biological data from the plasma of COPD patients and controls through a non-hypothesis-driven wide proteomic approach using tandem liquid chromatography–mass spectrometry. As a second step, a cluster analysis was applied to the proteomic results, and only at the end were these blindly generated clusters confronted with those clinical conditions that are the diagnostic targets of the study: COPD and AECOPD.

## 2. Materials and Methods

### 2.1. Study Design and Ethics

This is a case–control (COPD patients vs. healthy controls) as well as a case–case (patients in stable conditions vs. those with AECOPD) study, prospectively carried out in our hospital (and associated research center) to identify blood proteomic biomarkers that could be potentially used to identify individuals with COPD in the general population or differentiate AECOPD from periods of stability. The local Clinical Research Ethics Committee (CEIC) of our institution approved the study protocol (ref. 2014/5895/I), and the investigation was performed in close accordance with the principles of the Helsinki Declaration. All participants provided their informed written consent after receiving complete information on the objectives and techniques included in the study.

### 2.2. Study Population

Stable COPD patients were recruited from our monographic outpatient clinic, whereas AECOPD patients were obtained from the hospital ward of the respiratory department. Healthy individuals were selected through a medical questionnaire as well as normal blood analysis and forced spirometry. Individuals were carefully preselected in a random process from age and sex subgroups, ensuring maximum intergroup age- and sex matching; and all of them were European Caucasians from the Mediterranean area. Some individuals included in the present investigation have also participated in a wider multicentric project aimed at the identification of biomarkers for different COPD phenotypes following comparisons between predefined classical clinical groups [14,15]. Diagnosis of COPD and its degrees of severity occurred following current guidelines [4,16]. Briefly, criteria for COPD diagnosis were based on smoking history and a post-bronchodilator FEV_1_/FVC < 0.7, while the disease severity was assessed both through (a) the level of FEV_1_ impairment (GOLD I–IV), and (b) this item plus symptoms and an earlier history of AECOPD (GOLD A–B, E). [4] Forced spirometry was carried out following ATS-ERS standards [17], and reference values were those for a Mediterranean population [18,19]. Clinical stability (SCOPD) was defined as the absence of AECOPD in the previous three months, whereas AECOPD was designated as the presence of a sudden worsening of respiratory symptoms requiring additional therapy [4], and both conditions were certified by a senior specialist in respiratory medicine. This care team was independent of the research group that performed the proteomics study. Patients who were associated with other chronic respiratory disorders were excluded.

### 2.3. Clinical and General Data

Demographic and clinical data were obtained from both (a) clinical records including image techniques, blood analysis, and lung function tests, and (b) a standardized questionnaire on respiratory symptoms.

### 2.4. Blood Sample Collection

Peripheral venipuncture was used to obtain blood samples, which were then put into tubes with K3-EDTA for plasma analyses. All tubes were centrifuged for 15 min at 1500× *g*. and supernatants were transferred to fresh tubes and kept at −80 °C until analysis.

### 2.5. Label-Free Liquid Chromatography/Tandem Mass Spectrometry (LC–MS/MS)

Label-free quantification (LFQ) proteomics was performed using an EASY-nLC 1000 and an LTQ-Orbitrap Fusion Lumos mass spectrometer (Thermo Fisher Scientific, Waltham, MA, USA). Full information on sample preparation, instrument parameters, protein identification, and quantification procedures has been published elsewhere ([20] and its on-line supplement).

### 2.6. Statistical Analysis and Data Processing

Sample size calculation: The sample size for the study was estimated based on some of our previous studies and using the PC-size software GRANMO 7.10, 2010; https://laalamedilla.org/Investigacion/Recursos/granmo.html [20,21,22]. We assumed an 80% power to detect differences of more than 20% in primary outcomes, with a level of significance (*p*) equal to or less than 0.05.

Clinical data: Descriptive statistics on demographic and clinical data are presented as mean ± standard deviation or proportions. All continuous variables were examined for normal distribution with the Kolmogorov–Smirnov test and variance homogeneity with the Levene test. Since all clinical variables showed normal distribution, independent samples *t*-tests were used to compare groups pairwise, whilst Pearson’s coefficient was employed to evaluate potential correlations among quantitative variables. Categorical variables were analyzed pairwise with the Fisher exact test. Bonferroni *p*-value adjustment was applied when multiple comparisons were present. An adjusted *p*-value ≤ 0.05 was again judged as statistically significant.

### 2.7. Proteomic Data Preprocessing

Data normalization: To lessen the skewness of variable distributions and make them available for parametric analysis, protein quantitation values were log_2_ transformed.

Immunoglobulin standardization: LC–MS/MS divides immunoglobulins into constant chain and/or complete protein codes. So, a unifying method that combines both codes into a unique classification was applied to ensure the consistency of data. The sum of raw LFQ values was subsequently conducted for these new groups, resulting in the generation of “normalized datasets”.

Data filtering: We calculated the proportion of missing values in Log_2_ normalized raw proteomics datasets for each protein and discarded those proteins missing in more than 10% of samples.

Missing Value’s imputation: After removing proteins with >10% of missing values, SampMin imputation was used on the already filtered datasets [23]. These imputed values were calculated following the formula 0.95*min-value and were used for further steps of the statistical analysis.

Generation of sub-datasets for cluster analyses: The following two sub-datasets were generated for cluster analyses based on raw data: the dataset for COPD diagnosis included data from controls and SCOPD groups, whereas the dataset for identification of exacerbations included data from SCOPD and AECOPD subgroups.

### 2.8. Biological Cluster Generation and Evaluation

Machine-Learning unsupervised cluster generation: Clustering methods grouped study individuals based only on their proteomic profile. Scikit-learn Machine Learning module in Python (Wilmington, DE, USA) and K-means for either 2, 3, or 4 clusters, indistinctly, were generated, and a total of 5000 iterations was set to detect the best clustering [24]. We used the machine learning method proposed by Monti et al. to obtain the best informative and more robust number of K-means clusters, and this procedure also generated new ensemble clustering [25]. Finally, other complementary clustering methods, including Affinity propagation, MeanShift, and Agglomerative Clustering (i.e., Ward), were also tested [24].

Differences in general clinical characteristics between proteomic-based clusters: The general characteristics (age, sex, and BMI, among others) of the individuals included in the study were analyzed in each of the clusters obtained and selected in the earlier steps. As previously mentioned, we used the Fisher exact test to determine differences between categorical data, while independent samples *t*-tests were used to pairwise compare continuous data.

Evaluation of clinical composition of clusters: Following the above-mentioned blind approach, the obtained clusters were finally confronted with clinical groups. Based on coincidences or discrepancies of the former with the latter, a confusion matrix was generated and sensibility (SE), specificity (SP), Predictive Positive Value (PPV), Predictive Negative Value (PNV), Overall Detection Accuracy (Acc), and Matthews Correlation Coefficient (MCC, commonly known as the phi coefficient, φ, or r_φ_) were calculated for the most relevant clusters.

Analysis of the Proteomic Profiles of selected clusters: Once clusters had been blindly obtained, evaluated, and selected, the proteomic differences between them were analyzed. For that, the initially obtained “normalized datasets” were used again but the imputation of missing values was omitted in this particular case, and quantitative comparisons were limited to those proteins detected in at least 50% of individuals in each cluster. As previously mentioned, the homogeneity of the variances was determined by Levene’s test. When data met the homogeneity assumption, single pairwise comparisons were analyzed using the Student’s independent samples *t*-test, while the Welch *t*-test was used for data with unequal variances. We then calculated the protein fold change (log_2_FC). To consider the presence of multiple comparisons, the Benjamini–Hochberg procedure was used to calculate false discovery rates (FDR) [26], and only groups of peptides and/or proteins reaching values <0.05 were deemed to be of high confidence. Those groups of proteins/peptides differentially abundant between clusters (DAP) were mapped with UniProt (Uniprot, Geneve, Switzerland) via REST API, being limited to human proteins in the UniProtKB dataset (v2023) (Uniprot) [27].

Protein–protein interactions (PPI): The STRING platform (STRING Consortium, Kearney, NE, USA) was used (interaction score ≥ 0.4) to further evaluate the biological significance of DAPs, and, since this database’s coverage of immunoglobulin interactions is very uneven, the following additional databases were also employed: UniProt, tfact2gene (EXTRI, Herning, Denmark), Reactome-FIs (Cytoscape, Seattle, WA, USA), MPIDB MINT (Database Commons, Beijing, China), MatrixDB (Database Commons), iRefIndex, Int-Act, InnateDB, IMEx, HPIDb, EBI-GOA-nonIntAct (UCL-EBI, London-Hinxton, UK), ChEMBL (EMBL-EBI, Heilderberg-Hinxton, Germany-UK), and British Heart Foundation—University College London (bhf-ucl). The Cytoscape program (v3.8.0, http://www.cytoscape.org, accesed on 15 January 2024) was used to create network representations, which were then used to visualize combined interactions.

## 3. Results

### 3.1. Subject Characteristics

The final population was composed of 24 clinically stable COPD patients (SCOPD), 10 patients with an AECOPD, and 10 healthy individuals (control), and a summary of the most relevant clinical features in these groups is presented in Table 1. Concerning the demographic data, all the groups were comparable, and the two COPD groups were also comparable regarding their cigarette exposure, lung function, illness severity, and exacerbator profiles. Only the leukocyte and neutrophil counts were higher in the AECOPD group when compared with the other two groups, and the C-reactive protein (CRP) shows a strong tendency along the same lines. However, all these differences were nullified when new analyses were performed in the former group during the next phase of clinical stability.

### 3.2. Characteristics of the Detected Proteins

Overall, 256 peptide groups, which correspond to 361 single proteins/peptides, and 93 groups of immunoglobulin peptides, corresponding to 125 single peptides, were detected in the plasma of the study participants. The complete list of the identified proteins/peptides is available in the Appendix A.

### 3.3. Clinical-Blind Clusters under Evaluation

#### 3.3.1. COPD Diagnosis in a Stable Condition

To assess the potential of the clustering method for identifying the presence of COPD, a total of 11 clusters using SCOPD and control proteomic data were used (Appendix A). Twenty-seven of these clusters were then generated through a simple K-means analysis arranged to generate two, three, or four clusters (i.e., nine for each one). As previously mentioned, the best cluster number was chosen following Monti’s machine-learning approach, and it was two. 

Unfortunately, this clustering analysis was unable to approximate the segregation between COPD and healthy controls when proteomic data were finally confronted with these clinical groups (Table 2 and Table 3). The accuracy was low, reflecting poor sensitivity and even worse specificity [28].

#### 3.3.2. Identification of Exacerbation Episodes

The same general procedure was performed to identify the acute episodes, but this time data from AECOPD and SCOPD were used. The best cluster number was again two. However, on this occasion, the proteomic clustering was able to detect most of the patients with exacerbation, clearly differentiating them from those in a stable condition, with a high degree of accuracy (79.3%) (Table 4 and Table 5). Indeed, this two-cluster approach showed good specificity and sensitivity. The other algorithms that were also tested (Affinity Propagation, Mean Shift, and Ward Agglomerative Clustering) did not add significant improvements to the former results. 

### 3.4. Key Biological Processes Differentiating AECOPD from Clinical Stability According to Clinically Blinded Clusters

Since K-means-2 clusters accurately differentiated the exacerbated patients from stable ones, a deeper analysis was performed to identify the differential proteomic profile of both clusters. Cluster A (the one closer to an “exacerbator-like profile”) vs. Cluster B (a “SCOPD-like profile”) reported a total of 38 groups of DAPs, 13 over- and 25 under-represented in the former cluster. These results correspond to 54 proteins/peptides in the UniProt Database (14 would be over- and 40 under-represented) (Appendix A). The protein–protein interaction network analysis of those results revealed that the DAPs were associated with five biological processes, including the global inflammatory response, antibody-mediated immune mechanisms, blood coagulation, lipid profile modulation, and complement pathways (Figure 1 and Figure 2).

## 4. Discussion

The most relevant result of the present study is the demonstration that it is possible to blindly approximate the identification of COPD patients with an acute exacerbation, segregating them from those in stable conditions, by using only their systemic proteomic profile. Moreover, the entire approach was independent of clinical data. In other words, the differences observed between the stable and acutely exacerbated COPD came from a technique unbiased by any pre-existing hypothesis. Our findings not only enabled the obtention of new biomarkers for AECOPD but also may suggest additional pathophysiological mechanisms occurring in these acute episodes, thus potentially opening the way for new therapeutic strategies. To offer more personalized and precise medicine for COPD patients, a more accurate diagnosis and treatment are especially interesting [12,29]. Unfortunately, our methodological approach was unable to differentiate stable COPD patients from healthy individuals, which would have been of potential utility in the screening procedures.

The present findings reinforce our general hypothesis that a proteomic analysis blind to any clinical data may be useful to characterize certain circumstances linked to COPD, such as in the case of AECOPD. Some of the biomarkers obtained in the blood of exacerbated patients are relatively new since they do not fully correspond to those suggested in most of the earlier literature. Although the proteomic network disclosed two relatively expected groups (proteins/peptides related to inflammatory and antibody-mediated immune responses), they appeared along with other aggregations composed of the molecules involved in blood coagulation, the modulation of the lipid profile, and the complement system. Moreover, the approach used in the present study may also be useful to characterize different COPD phenotypes (chronic bronchitis, pulmonary emphysema, frequent exacerbator, eosinophilic inflammation, etc.). This possibility should be explored in further studies.

### 4.1. The Clinically Blind Approach

Many of the previous ‘*omic*’ studies carried out to identify the biological markers of COPD and/or its different circumstances have been performed through transcriptomic analyses since this approach enables the obtention of very broad results at a relatively low cost [30,31,32]. However, there is also a need to incorporate studies on proteins since these molecules are ultimately the actual performers in most biological processes. However, former proteomic studies have generally followed strategies that include different components linked to pre-existing hypotheses. On the one hand, some of these studies have been oriented to detect the markers that were predefined by the most widely accepted hypotheses on the pathophysiology of either COPD or AECOPD [9,10,33]. This is a restrictive bias that intrinsically limits the scope of the results [34]. In our study, the broadest search was performed blindly via untargeted protein screening using LC–MS/MS. The other non-blind strategy used in most of the previous studies consisted of directly comparing the biological results between well-predefined clinical groups (i.e., patients vs. healthy controls, or among different sets of COPD patients grouped by their clinical characteristics, phenotypes, or even treatable traits) [9,10,33]. This is a logical design to assess the differences between clear clinical profiles [12,35]. However, this latter strategy does not enable obtaining biological profiles that would be completely independent of the previous knowledge. Moreover, some of these studies have been based on pools of blood samples from patients considered to belong to the same clinical group, an approach that has an implicit bias that involves the loss of inter-individual information and avoids potential post-hoc segregation based on the biological results. In contrast, the approach used here consists of a protein-based cluster analysis, which did not consider any clinical data until the final step of the analysis, and then only investigated whether proteomic clusters make clinical sense. This is an approach that provides maximum independence to data, facilitating the obtention of novel findings [25,36,37].

Although the use of cluster analysis is not new in the field of COPD, in the vast majority of the already published studies, it has only been applied to clinical variables, including some co-authored by members of our group [38,39,40]. This strategy is fully justified to identify new clinical phenotypes, but it does not consider the contribution of the corresponding biological substrates (endotypes) and, therefore, that of their corresponding pathophysiological mechanisms.

The specific usefulness of cluster analysis to identify each one of the two targeted clinical populations of interest for the present study (i.e., COPD and its AECOPD) is analyzed in the following two sections.

### 4.2. Clinical Usefulness

#### 4.2.1. COPD Diagnosis

One of the main objectives of the present study was to assess the usefulness of our approach in detecting the presence of COPD. This could be useful to reduce underdiagnosis (considered to be around 70%) [41,42], especially in patients with mild to moderate disease, who are permanently in stable clinical conditions, when the symptoms may not be very suggestive of the disease. Unfortunately, the clusters obtained here only showed moderate-to-low levels of accuracy to identify the disease. This could be explained simply by the heterogeneity of COPD. In fact, due to its open design, our population included stable patients who combined different characteristics and phenotypes (males and females, with rare or more frequent exacerbations and the presence or absence of blood eosinophilia, etc.). These are groups that may show mildly differentiated proteomic profiles, as seems to be indicated in some previous studies [43,44]. Other authors, however, also using a case–control design and a similar proteomic methodology, have found a relatively common profile for stable COPD patients [12,45,46]. It is therefore possible that future cluster analyses including larger populations, with a high number of subjects from each demographic and phenotypic group, and/or incorporating more biological levels (RNA, metabolites, etc.), will enable the obtention of more promising results. Another explanation for our poor results in COPD diagnosis would be that the biological changes were of a very low intensity in the stable patients, which would make them partially undetectable through the methods employed here and their proteomic differences with the controls.

#### 4.2.2. Identification of AECOPD

The acute episodes were identified through the cluster analysis used in the present investigation. This is the first proteomic study that enables such an identification, which can be potentially useful for both clinical practice and the more objective and precise inclusion of AECOPD patients in clinical trials. The current definition of AECOPD is based on worsening symptoms requiring modifications to the treatment. As previously mentioned, this has undoubted subjective components both on the side of the patient and from the perspective of the care team.

#### 4.2.3. Main Protein Groups Differentiating Exacerbations from Stability According to Biological Clusters

When the differentially expressed proteins were analyzed among the clusters obtained in the present study, they appeared to be concentrated into five large groups (Figure 1). Two of them were certainly expected from the beginning since they were those belonging to inflammation and the antibody-mediated immune response [3,4,7]. Indeed, the inflammatory proteins were generally overexpressed in the cluster linked predominantly to AECOPD, indicating the already well-known increase in inflammatory activity occurring during these acute events [4]. These proteins included some that have already been described to be increased in AECOPD, as is the case for Serum Amyloid A1 (SAA), Haptoglobin (HP), Lipopolysaccharide binding protein (LBP), and other markers related to inflammation [12,47,48,49]. In particular, SAA is known to induce the activation and chemotaxis of neutrophils and other inflammatory cells, promoting the release of matrix metalloproteases and inducing the expression of proinflammatory cytokines [50]. The proteins/peptides present in the second group, in turn, were mostly immunoglobulin fractions, which were decreased in the cluster most closely associated with AECOPD. These fractions were free light chains (Kappa and Lambda), a finding that can be interpreted in two different ways. It may be due to a reduction in their concentration, secondary to the increased use during AECOPD episodes, or it may have been the result of a synthesis failure in different immunoglobulins, which in turn could have facilitated the acute episodes. This latter hypothesis appears to be more feasible since various authors have reported low blood levels of free light chains of immunoglobulins in COPD patients, both at AECOPD or even during stable conditions [51,52,53]. In the following paragraphs, the other three protein groups that were associated with the cluster most closely linked to AECOPD will be analyzed in more detail.

A third group was constituted by proteins participating in blood coagulation, which is not entirely surprising. In fact, a prothrombotic environment is not rare in COPD patients [54,55], especially during AECOPD, where some additional factors such as a greater reduction in physical activity are also added [56,57,58]. Our present proteomic findings suggest that the specific prothrombotic status during AECOPD may be due to a specific pathway imbalance derived from the reduced activation of plasminogen as a result of the action of the histidine-rich glycoprotein (HRG) [59] and/or the activation of fibrinogen production (FGB/FGA) and further transformation to fibrin [60].

However, the major biological novelties found in the present study are related to the remaining two groups, the proteins/peptides participating in the lipid profile modulation or in the complement system. On the one hand, two proteins involved in lipid transport and metabolism (APOC1 and APOA2) were decreased in the cluster associated with AECOPD. This is in accordance with the results of a previous hypothesis-driven study on COPD [61], also being similar to what has been reported in other entities, such as severe infections and sepsis, where most of the lipoproteins, lecithin, and lipid transporters (such as cholesterol acyltransferase and plasma cholesteryl ester transfer protein) seem to be decreased due to the inhibition of the reverse cholesterol transport [62]. Even though the underlying mechanism is partially unknown in the particular case of COPD [63], it would be possible to speculate that it might be similar to that described for infections since this is the main cause of most AECOPD. Likewise, the α(2)-Heremans–Schmid glycoprotein (AHSG), which is related to obesity and a lack of activity [64,65], was also found to be decreased in this same cluster. It is known that obesity and reduced physical activity are frequently observed in COPD patients. Moreover, some members of our group also demonstrated the role of these two circumstances as risk factors for AECOPD [39,66]. In contrast, the zinc-α-2-glycoprotein (AZGP1) was found to be increased in that same protein cluster. This protein is involved in lipolysis and body fat loss and has been associated with tobacco smoking, although the latter is controversial [67,68]. It is interesting to highlight that the body weight and/or lean mass loss become enhanced during AECOPD, a factor that is associated with patients’ vital prognosis [56,69,70].

The present findings in some of the proteins of the complement system (C3, C5, and the C4BPA inhibitor) also stand out among our results and are in close agreement with one of our previous articles [20]. This system directly promotes bacterial lysis, is involved in the opsonization of pathogens, and contributes to washing out immune complexes from the blood. The activation of the complement system occurs through three interconnected pathways: the classical (bound to the antigen–antibody complex), the lectin pathway (involving the recognition of surface carbohydrates), and the so-called ‘alternative pathway’ (started when the C3b protein binds a microorganism, and with a relatively low level of activity) (Figure 2). These three pathways concur to sequentially produce C3 and C5 convertases, and the membrane attack complex (MAC) that induces cell lysis by generating gaps in the membrane [71,72,73]. In the present study, an overrepresentation of the α chain of the C4b-binding protein (C4BPA), which is an inhibitor of the C3 convertase in the classical and lectin pathways but not in the alternative one [74], was also observed. The parallel increase in C3 and C5 (the initial products of the two main reactions of the complement cascade) together with that of its inhibitor (C4BP) would indicate a decrease in the global efficiency of the activation of the complement system during AECOPD. Our results suggest three possible interpretations and consequences for these findings. First, this, along with a probably diminished humoral immune response (evidenced by the previously discussed coexistence of a reduction in immunoglobulins), would lead to compromising the host’s defenses against infections. Second, the increase in C3 may also reflect the greater activation of the alternative pathway as a possible compensatory effect. Finally, our findings in C3 and C5 could alternatively be the consequence of the action of certain pathogens to evade the complement system. It is worth noting that some of the previously published hypothesis-driven studies have found increases in C7 and C9 (later participants in the MAC activation) in AECOPD [20,75], a particular finding that would agree with any of the three above-mentioned possibilities. Therefore, the clarification of this specific point merits further and more specific investigations.

In contrast, the simultaneous absence of the overexpression of the β chain of C4BPA could be interpreted as a reflection of an actual increase in the α7β0 isoform. It is well-known that the latter increases in the presence of inflammatory stimuli, inducing a state of tolerance to noxae in certain dendritic cells [76,77]. Interestingly, it has been reported that various bacteria, including Streptococcus pyogenes, recruit C4BP to evade the immune system [78].

### 4.3. Strengths and Limitations

The main strength of the present study is its completely blind approach to clinical conditions: in other words, a non-biased approximation to the diagnosis of either a very prevalent chronic respiratory disease or its acute episodes of clinical impairment. Although it failed in the primary objective, it was accurate for the latter.

A secondary strength is more technical. We used K-means to generate the clusters since this approximation enables a global analysis of the proteome without considering any pre-hoc selection of proteins based on criteria such as their variability or potential relevance. This, by contrast, would have been the case for some classical alternatives such as principal component analysis (PCA) or partial least-squares regression analysis (PLS). PCA “optimizes” data as it reduces protein values’ dimensionality, transforming them into principal components and focusing more on those with the higher overall variability but ignoring the individual influence of either the magnitude or relative correlations of each element. In turn, the PLS approach builds the results around an initially selected variable [79,80].

The study also has some potential limitations. First of all, the sample size is relatively small, although it is appropriate for the objectives of the study and similar to other previously published proteomic studies [35,61]. The single-center character of our design can be viewed as both a limitation and a strength since it ensures the strict homogeneity in the inclusion–exclusion criteria for the two different COPD groups, a point that is especially relevant in the case of AECOPD for the potential bias discussed above. Moreover, our results do not differ substantially from some of the previous hypothesis-driven studies carried out by our group with a multicenter design [20]. 

Certainly, the observational and cross-sectional design of the present study does not enable the drawing of any causal inferences between the proteomic findings and clinical conditions. However, it suggests novel pathophysiological hypotheses that should be tested in the future with complementary designs.

Moreover, although it is worth noting that this study provides relevant insights into potential proteomic profiles for Caucasian COPD patients during exacerbations and clinical stability, further studies should test their validity in other populations. In any case, and according to the estimations of a recent publication, our findings may be especially useful for what constitutes a significant fraction of the global COPD population [81].

Finally, we wish to mention that the entire proteome spectrum has not been explored in the present study. Since we did not deplete the most abundant plasmatic proteins, those with the lowest abundance became lost. However, the alternative technical option of depleting these abundant proteins would have resulted in the loss of many components of the humoral immune response and the complement system. Nevertheless, we have tried to overcome this specific limitation by adding some of these proteins to a post-hoc analysis (data not included in the present paper but in the Appendix A). For this, we employed multiplex panels containing most of the more accepted protein candidates to be linked to AECOPD but with presumably low concentrations in the blood. However, this did not improve the accuracy of our blind prediction either for COPD nor for AECOPD.

## 5. Conclusions

The most relevant conclusion of the present study is that the use of a completely blind methodology enables the reasonable identification of patients with AECOPD through a massive proteomic analysis of their blood samples. In addition, it has made it possible to complement previous studies carried out with alternative designs, suggesting new pathophysiological pathways related to these acute episodes. Indeed, it has not only confirmed the involvement of inflammation and changes in humoral immunity in AECOPD, as were quite predictable, but also modifications in the coagulation pathways and the regulation of lipid metabolism and the complement system. On the contrary, our method added no new elements for the screening of COPD.

## Figures and Tables

**Figure 1 cells-13-00866-f001:**
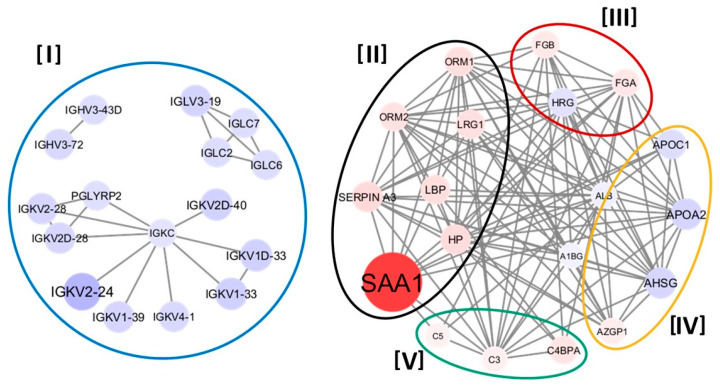
Protein–protein interaction network of differential abundant proteins (DAPs) in the exacerbation diagnostic K-means-2 clusters obtained with LC–MS/MS. Legend: PPI networks performed on DAPs between Cluster A vs. Cluster B obtained in K-means used for identification of the exacerbation profile. Each node lists the gene name of the identified proteins. The size and shade of the nodes represent the median log2 fold difference of DAPs, with red and blue showing over- and under-represented proteins, respectively. Edges stand for already known and predicted protein interactions obtained from the STRING database, as well as the other public databases that were selected for the study. DAPs are grouped into proteins related to [I] humoral immunity, [II] inflammation (includes cytokines, chemokines, and acute-phase proteins), [III] coagulation, [IV] lipoprotein particle remodeling systems, and (V) the complement system.

**Figure 2 cells-13-00866-f002:**
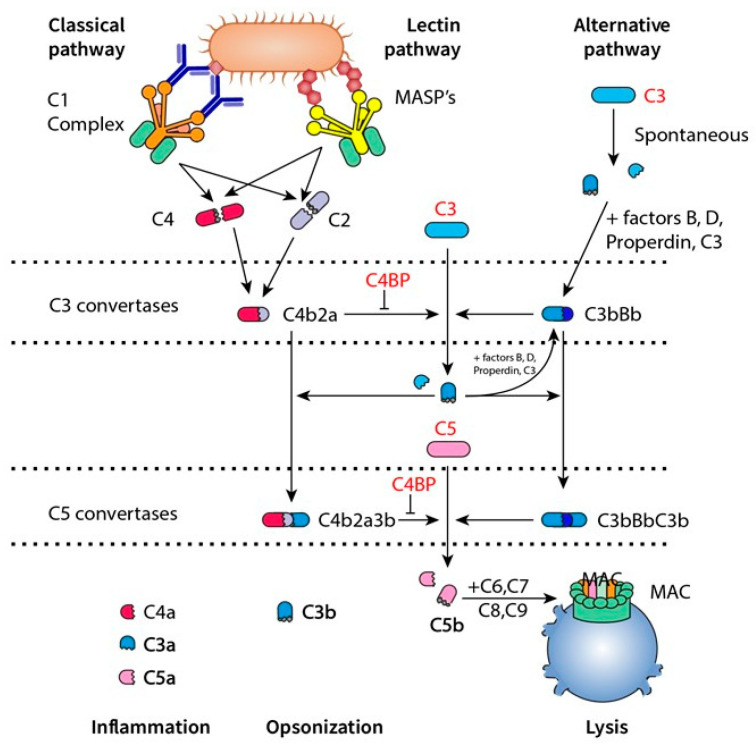
Proteins acting in the three pathways of the complement system in COPD patients. Legend: those proteins overexpressed in the cluster with the higher number of exacerbated patients appear in red. MASP: MBL-associated serine protease.

**Table 1 cells-13-00866-t001:** Clinical characteristics of the study population.

		COPD	
	CONTROL (n = 10)	SCOPD (n = 24)	AECOPD (n = 10)
**General characteristics**			
Age, yr.	63 ± 11	66 ± 9	63 ± 7
Males, % in the group	60	58	56
BMI, kg/m^2^	25.3 ± 2.0	24.8 ± 6.9	26.2 ± 4.7
**Smoking status**			
Current or Ex, % in the group	60	100 *	100 *
Pack/years smoking	13.2 ± 15.5	52.8 ± 22.8 ***	62.0 ± 34.2 **
**Lung Function**			
FEV_1_, % pred.	82 ± 3	40 ± 10 ***	36 ± 12 ***
FEV_1_/FVC, %	80 ± 2	49 ± 10 ***	52 ± 12 **
DLco, % pred.	NA	45 ± 14	40 ± 8
**GOLD Stage**			
I–II, % within the group	---	21	22
III–IV, % within the group	---	79	78
A–B, % within the group	---	25	20
E, % within the group	---	75	80
**Exacerbations**			
last year, n	---	2.6 ± 2.6	2.4 ± 1.6
0–2/year, % in the group	---	42	50
>2/year, % in the group	---	58	50
**Conventional Blood Analysis**			
Leucocytes, n/µL	7216 ± 1356	8384 ± 2693	12,486 ± 5551 *^,#^
Neutrophils, n/µL	4337 ± 1030	5409 ± 2323	10,243 ± 5665 *^,##^
Eosinophils, n/µL	221 ± 142	214 ± 230	119 ± 181
CRP, mg/dL	0.4 ± 0.2	0.9 ± 1.4	3.1 ± 3.9
Fibrinogen, mg/dL	207 ± 31	210 ± 57	221 ± 65

Values are shown as mean± SD or percentages. Significances (*p*-adjusted values): *, *p*  <  0.05; **, *p* < 0.01; ***, *p* < 0.001 for COPD compared to control; ^#^, *p* <  0.05; ^##^, *p* <  0.01 for SCOPD compared to AECOPD. Abbreviations: SCOPD, stable COPD; AECOPD, exacerbated COPD; BMI, body mass index; FEV_1_, forced expiratory volume in the first second; FVC, Forced Vital Capacity; DL_CO_, diffusion capacity for carbon monoxide; CRP, C-reactive protein; NA, non-available.

**Table 2 cells-13-00866-t002:** Main clinical characteristics in each K-means-2 cluster found by proteomics, and confrontation of these clusters with the distribution of actual COPD and control groups.

	Proteomic Clusters	
	A	B
**Individuals, n**	24	10
**General characteristics**		
**Age, yr.**	64 ± 9	67 ± 10
**Males, n (% in the cluster)**	12 (50)	8 (80)
**BMI, kg/m^2^**	25.7 ± 6.2	23.1 ± 4.8
**Group**		
**CONTROL, n (% in the cluster)**	8 (33)	2 (20)
**SCOPD, n (% in the cluster)**	16 (67)	8 (80)

Values are expressed as mean ± SD, or percentage. Abbreviations: BMI, body mass index; SCOPD, stable COPD.

**Table 3 cells-13-00866-t003:** Clustering outcomes for COPD diagnosis.

N of Clusters	SP	SE	PPV	PNV	ACC	MCC	Raw *p*-Value	Bonferroni
2	20 (16)	67 (19)	67 (19)	20 (16)	53 (20)	−0.13	0.68	1

Values are expressed as percentage (CI95). Cross-validation Fisher and Bonferroni *p*-values are included. Abbreviations: SP, specificity; SE, sensitivity; PPV, Predictive Positive Value; PNV, Predictive Negative Value; ACC, accuracy; MCC, Matthews correlation coefficient.

**Table 4 cells-13-00866-t004:** Main clinical characteristics in each K-means-2 cluster and confrontation with the distribution of stable and exacerbated COPD patients.

	Proteomic Clusters	
	A	B
**Individuals, n**	13	21
**General characteristics**		
**Age, yr.**	66 ± 8	64 ± 9
**Males, n (% in the cluster)**	6 (46)	13 (62)
**BMI, kg/m^2^**	29.3 ± 7.4	23.6 ± 4.8 *
**COPD group**		
**SCOPD, n (% in the cluster)**	5 (39)	19 (91) **
**AECOPD, n (% in the cluster)**	8 (61)	2 (9) **

Values are expressed as mean± SD, or percentage. Significance: *, *p*  <  0.05; **, *p*  <  0.01 B compared to A. Abbreviations: AECOPD, acutely exacerbated COPD; SCOPD, stable COPD; BMI, body mass index.

**Table 5 cells-13-00866-t005:** Clustering outcomes for identification of exacerbations.

N of Clusters	SP	SE	PPV	PNV	ACC	MCC	Raw *p*-Value	Bonferroni
2	79 (25)	80 (25)	62 (30)	91 (18)	79.3 (25)	0.55	˂0.01	˂0.01

Values are expressed as percentage (CI95). Cross-validation Fisher *p*-value and Bonferroni correction value are shown. Abbreviations: SP, specificity; SE, sensitivity; PPV, Predictive Positive Value; PNV, Predictive Negative Value; ACC, accuracy; MCC, Matthews correlation coefficient.

## Data Availability

Data are contained within the article and Appendix A.

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
