# Peer review of "Proteomic Blood Profiles Obtained by Totally Blind Biological Clustering in Stable and Exacerbated COPD Patients"

_cells, 2024, doi:10.3390/cells13100866_

Round 1

Reviewer 1 Report

Comments and Suggestions for Authors

The authors examine protenomic signatures - finding differences between Stable COPD, AECOPD, and "normal" non COPD patients.

Comments

PCR ( Table 1) needs to be spelled out.

The "stable COPD" had more exacerbations in the prior year than the AECOPD group-- which is a bit odd.  Were there any "Stable COPD" with no prior year exacerbations?  That would be a better comparator group ( and a true Stable COPD group)

The Current GOLD classification is ABE, not ABCVD.  This needs to be changed.

The Leukocyte count was twice as high in the AECOPD group relative to the other two.  Is it possible that many of the findings are related to this?  Can this be looked at?  Similar question for the PCR.

Author Response

The authors examine proteomic signatures - finding differences between Stable COPD, AECOPD, and "normal" non-COPD patients.

   Thank you for your kind review, where you have considered our article positively, and for the comments that you have indicated for improvement.

Comments

PCR (Table 1) needs to be spelled out.

   It has been corrected to CRP and its spelling incorporated into the new version.

The "stable COPD" had more exacerbations in the prior year than the AECOPD group-- which is a bit odd.  Were there any "Stable COPD" with no prior year exacerbations?  That would be a better comparator group (and a true Stable COPD group)

   This reviewer's suggestion is very interesting to discuss, and we will try to explain our chosen options. On the one hand, we would like to emphasize that the number of exacerbations in the previous year was similar between the SCOPD and AECOPD groups (see table 1), although some of the members of the former group showed a significant trend of having more than those in the latter (therefore, being members of the widely accepted ‘frequent exacerbator phenotype’). Our group tried to include diverse profiles of stable COPD patients in the SCOPD group, with more or fewer exacerbations in the previous year (see the SD in the table) to have a more representative population. Moreover, we chose to include SCOPD patients with similar levels of severity to that of those individuals who were exacerbated in the moment of obtaining the blood, since otherwise, we would have been comparing different phases of the chronic disease. This would have been the case if we had included patients with a total absence of acute episodes, who generally show mild-to-moderate disease. Moreover, we made sure that patients in the SCOPD group were completely stable, meeting the strict criteria that are common to similar studies (3 months without changes in symptoms or medication). An important reason for the different aspects of our design was to obtain proteomic footprints with real clinical usefulness regarding the diagnosis of the disease and/or the exacerbation.

   As mentioned by the reviewer, in a previous targeted non-blinded and wider study, we found that the footprint of exacerbated patients was partially similar to that shown by the exacerbator phenotype during clinical stability, very far from that of non-frequent exacerbators [reference #19 of the manuscript). However, in the present manuscript we address a more practical clinical question: Is it possible to differentiate the proteomic footprint of patients with an actual episode of AECOPD from those with a similar COPD severity but considered by expert clinicians as totally stable?

The Current GOLD classification is ABE, not ABCVD.  This needs to be changed.

We agree with the Reviewer’s comment, and this has been modified accordingly (Table 1), and the quotes updated to the new GOLD 2024 recommendations.

The Leukocyte count was twice as high in the AECOPD group relative to the other two.  Is it possible that many of the findings are related to this?  Can this be looked at?  Similar question for the PCR.

   Yes, the reviewer is right. These blood findings, which, from a clinical point of view, are characteristic of AECOPD (higher values of leukocytes and CRP) may have influenced the proteomic results. This is part of the assumed effects of these acute episodes, widely considered as being mainly the result of airway infections, on its proteomic footprint.

   However, this was not the case for patients included in the AECOPD group when they recovered from the acute episode and were in a subsequent phase of clinical stability. Then, data were as follows: Leukocytes 7768±1384 n/μL, Neutrophils 4997±984 n/μL, and CRP 0.7±0.9 mg/dL.

Reviewer 2 Report

Comments and Suggestions for Authors

The study "Proteomic Blood Profiles Obtained by Totally Blind Biological Clustering in Stable and Exacerbated COPD Patients" by Cesar Jessé Enríquez-Rodríguez et al. presents a novel and commendable approach by employing a blind clustering method for identifying potential biomarkers in COPD. This innovative technique, especially the use of label-free liquid chromatography/tandem mass spectrometry combined with Scikit-learn machine learning for clustering, showcases the authors' commitment to advancing research methodologies in the field. Such approaches are pivotal for the unbiased discovery and characterization of disease-specific proteomic profiles. The findings, particularly the potential of the blinded approach to segregate AECOPD from stable conditions, have significant implications for personalized medicine and clinical trials. This work potentially paves the way for more targeted and effective interventions for COPD patients.

The authors have meticulously prepared this article. Overall, the paper is well-prepared and written. However, the author should address the following issues, which could help improve the significance of the study:

·       The relatively small sample size and the study's single-centre nature may limit the generalizability of the findings. Future studies exploring a broader demographic and phenotypic spectrum of COPD patients could enhance the robustness and external validity of the research. 

·       The authors must provide demographic data and a Consort flow chart.

Author Response

Comments and Suggestions for Authors

The study "Proteomic Blood Profiles Obtained by Totally Blind Biological Clustering in Stable and Exacerbated COPD Patients" by Cesar Jessé Enríquez-Rodríguez et al. presents a novel and commendable approach by employing a blind clustering method for identifying potential biomarkers in COPD. This innovative technique, especially the use of label-free liquid chromatography/tandem mass spectrometry combined with Scikit-learn machine learning for clustering, showcases the authors' commitment to advancing research methodologies in the field. Such approaches are pivotal for the unbiased discovery and characterization of disease-specific proteomic profiles. The findings, particularly the potential of the blinded approach to segregate AECOPD from stable conditions, have significant implications for personalized medicine and clinical trials. This work potentially paves the way for more targeted and effective interventions for COPD patients.

The authors have meticulously prepared this article. Overall, the paper is well-prepared and written.

   We greatly appreciate your thoughtful review and are honoured that you had such a favourable opinion of our manuscript. We agree that blind segregation of exacerbated stable patients may have relevant implications for diagnosis in both clinical and research settings.

However, the author should address the following issues, which could help improve the significance of the study:

  • The relatively small sample size and the study's single-centre nature may limit the generalizability of the findings. Future studies exploring a broader demographic and phenotypic spectrum of COPD patients could enhance the robustness and external validity of the research. 

   We only partially agree with this comment regarding the sample size and the single-centre nature of our study, for although we recognize the limitations implicit in a reduced sample size, this was calculated from one of our previous studies (reference #19), being also consistent with other earlier investigations in the field. Even though it may be considered a relatively restrictive approach it is fully justified considering the difficulties in obtaining blood samples of AECOPD patients before any treatment with steroids or other drugs that can modify the pathophysiological response of the body, as well as the high costs derived from LC-MS/MS proteomic analyses.

   Regarding the single-centre design of the present study, this decision was made to also obtain strength since it allowed us to have a greater integrity and homogeneity of individual data, with a more precise final comparison with clinical groups. With this approach, we avoided possible inter-hospital or inter-physician variations in patient classification, which could have affected the quality of our results, especially given the sample size.

   Certainly, there are also other omics studies with a multi-centre design and a much larger sample size. However, one of the approaches used in some of them is the use of sample pools, an option that has two important limitations. The first is the loss of inter-individual information that would limit segregation of patients. The second, and more serious limitation, would be that a priori classification involves the inclusion of each patient in one pool or another, which constitutes an implicit bias. Another criticism of wider studies is that most of them included centres from different countries with heterogeneity derived from their different health systems and the ethnicity of the patients. By contrast, all our patients came from a homogenous care system and professional care team (who were independent of the research group), and all of them were Caucasian from Southern Europe. Again, this can also be considered an advantage in terms of homogeneity and potential replicability in a similar environment, but it also limits the generalization of the results to other populations. Therefore, it would be important to validate the study in different populations. Finally, a broader (and more clinically heterogeneous) study probably could have masked several details that we identified. We consider that general approaches go against the purposes of a more personalized medicine. We have avoided a detailed criticism of the limitations of all different approaches used in previous articles since this would have led the Discussion outside the focus scope of the present manuscript. However, we already had and have also added different paragraphs supporting these ideas in the present version of the Discussion within the manuscript.

   In any case, we would like to emphasize that our research focused on a novel approach using a completely blind strategy and the use of artificial intelligence techniques, a methodology that, to our knowledge, has not been explored at all or perhaps slightly in the field of omic footprints in different profiles of COPD patients. We hope that further research will validate and expand our findings, thus contributing to the advancement of knowledge in this field.

  • The authors must provide demographic data …

   The main demographic data of our patients were or have been included now in the updated version of the manuscript.

The authors must provide … a Consort flow chart.

Even though we appreciate this comment we feel that a CONSORT diagram would not be appropriate in this case since the study is not a randomized clinical trial but rather a case-control and case-case study. Therefore, the manuscript has followed the guidelines of the STROBE guide. As already mentioned, patient selection was conducted among the population receiving health care at our centre, and given the expensive nature of the proteomic technique, the minimum sample size was calculated based on a previous multicentric study. The subsequent selection was carried out in a semi-random manner (AECOPD were consecutively recruited and ‘age and sex boxes’ were filled to match samples from people from the other two groups (Control and SCOPD). Moreover, before proteomic analysis, demographic similarity was again checked to ensure that the interindividual differences except clinical condition (i.e. health, clinical stability or exacerbation), would not influence our findings.

Reviewer 3 Report

Comments and Suggestions for Authors

The study of “The Proteomic Blood Profiles” reported in this paper is of interest. The reporting of the results could be presented with greater clarity.  The commentary of methodology and the results are often obscured by the complexity of language which obscures the meaning.

It seems advisable to divide this present submission into a brief report of “The Proteomic Blood Profiles” and a review or commentary on “Blind Biological Clustering” of data related to exacerbation in COPD patients.

Comments on the Quality of English Language

The commentary of methodology and the results are often obscured by the complexity of language which obscures the meaning.

Author Response

The study of “The Proteomic Blood Profiles” reported in this paper is of interest.

   We appreciate the Reviewer’s consideration of our results as interesting and worthy of publication.

The reporting of the results could be presented with greater clarity.  The commentary of methodology and the results are often obscured by the complexity of language which obscures the meaning.

   Following the Reviewer’s recommendations, we have tried to order the different sections better both in the Methodology and the presentation of the Results, with clearer sentences and paragraphs to facilitate reading and understanding for a wide spectrum of readers, from more basic scientists to experts in bioinformatics analysis and/or clinicians.

   We have updated and completed the references. However, we have only increased their number slightly since we already have 80!, including most of the more relevant ones.

It seems advisable to divide this present submission into a brief report of “The Proteomic Blood Profiles” and a review or commentary on “Blind Biological Clustering” of data related to exacerbation in COPD patients.

   We completely agree with the Reviewer. It is important to highlight that our research focused on two aspects: (1) a novel approach using artificial intelligence techniques in a completely blind statistical strategy. A methodology that, as far as we are aware, has not been explored at all or maybe slightly in the analysis of omic data in COPD patients; and (2) a non-hypothesis-biased wide exploration of the proteomic footprint of healthy controls and different profiles of COPD patients using untargeted LC-MS/MS. We have chosen to dissert the novelties of our research in the same order in the Discussion section since we feel this is much more progressive and clearer but we will change it if the reviewers wish.

   Therefore, this study sought to categorize subjects independently of the clinical data that physicians (who were independent from the present research team) had previously assigned to them, letting the wide biological spectrum of proteins/peptides classify the patients. Only in the second step, when clusters were finally confronted by clinical groups, did we seek to know what proteins this blind approach had used to make this apparently useful categorization. To do so, we explored the differences between the obtained protein categories.

   In line with this particular comment of the Reviewer we have now tried to better categorize the Discussion in the different aspects of the blind approach used in the study, and the actual footprint more characteristic of AECOPD patients.

Comments on the Quality of English Language

   We have carefully reviewed the new version of the manuscript with an expert in scientific and medical language, whose mother tongue is English (see the acknowledgements).

The commentary of methodology and the results are often obscured by the complexity of language which obscures the meaning.

   Indeed, as previously mentioned, we have tried to better sectorize the Methodology, Results and especially the Discussion of our results into two clear blocks (i.e. the blind approach and the specific proteomic footprints). By doing so we believe that the manuscript is much clearer in making it more evident that we are always talking about biologically-only generated clusters and their relevant proximity to actual clinical characteristics in COPD patients.

Round 2

Reviewer 1 Report

Comments and Suggestions for Authors

Prior comments have been addressed

Reviewer 3 Report

Comments and Suggestions for Authors

revised manuscript makes it suitable for publication.